# Estimated Pulse Wave Velocity (ePWV) in Different Glaucoma Types

**DOI:** 10.3390/biomedicines13082033

**Published:** 2025-08-21

**Authors:** Marija Bozic, Vesna Maric, Vladimir Milutinovic, Margita Lucic, Jelena Vasilijevic

**Affiliations:** 1Faculty of Medicine, University of Belgrade, University Clinical Center of Serbia, University Eye Hospital, 11000 Belgrade, Serbia; vesnamaric21@gmail.com (V.M.); bkjelena@gmail.com (J.V.); 2University Clinical Center of Serbia, University Eye Hospital, 11000 Belgrade, Serbia; dr.vladimir.milutinovic@gmail.com; 3University Clinical Center of Serbia, Center for Anesthesiology and Resuscitation, 11000 Belgrade, Serbia; margita.84@gmail.com

**Keywords:** glaucoma, arterial stiffness, pulse wave velocity

## Abstract

**Background/Objectives**: This study aimed to evaluate estimated pulse wave velocity (ePWV) in different glaucoma types. **Methods**: This was observational, cross-sectional, non-interventional study conducted on 127 primary open-angle glaucoma (POAG) patients, 59 primary angle-closure glaucoma (PACG) patients, 34 pseudoexfoliative glaucoma (PEX) patients, and 55 normotensive glaucoma (NTG) patients (total of 275 glaucoma patients). The control group (CG, 92 patients) consisted of patients with cataract. ePWV was calculated by the formula that was recommended by the Reference Values for Arterial Stiffness Collaboration from data on age and mean arterial blood pressure. The obtained results were processed by applying methods of descriptive (arithmetical mean, standard deviation) and analytical statistics, and comparisons of tested variables were performed using ANOVA. A p value less than 0.05 was considered statistically significant. **Results**: Statistically significant differences were found between patients with POAG and the CG (*p* value 0.042), and between those with NTG and the CG (*p* value 0.001). There was a statistically significant difference in ePWV values when comparing all tested patients with glaucoma and the control group (*p* = 0.001). **Conclusions**: Estimated pulse wave velocity may be a helpful tool in future risk assessment models for glaucoma.

## 1. Introduction

Glaucoma is increasingly recognized as a group of neurodegenerative diseases with multifactorial origins, which can culminate in irreversible damage to the optic nerve and potential blindness. Over the past 10–20 years, significant advancements have been made in understanding the pathophysiological mechanisms underlying glaucoma, leading to evolving definitions of the disease. Historically, the mechanical theory of glaucoma dominated scientific discourse, but vascular-related damage has garnered attention, presenting complex and inadequately understood pathophysiological mechanisms. Numerous studies suggest that vascular factors are implicated not only in the majority of glaucoma cases but are especially relevant in instances where the condition’s onset and progression cannot be solely attributed to elevated intraocular pressure (IOP), such as normotensive glaucoma.

Apart from local specificities, we must consider the eye as a part of the global vascular system, which is why research continues to delve into how systemic vascular factors influence the hemodynamics and hydrodynamics of the eye. One emerging hypothesis concerning the systemic vascular influence on glaucoma posits that alterations in ocular blood flow may stem from increased arterial stiffness—a phenomenon characterized by the decreased elasticity of larger arteries, including the aorta. Arterial stiffness can be assessed through various methodologies, one of which is estimated pulse wave velocity (ePWV). The recent literature has suggested ePWV as a promising new marker for mortality and cardiovascular risk [1]. While some studies [2,3] have investigated the relationship between arterial stiffness and glaucoma, there remains a notable gap in the literature regarding a comparative analysis of ePWV values across different types of glaucoma. Therefore, our study aims to address this deficiency by systematically examining ePWV levels in various glaucoma subtypes, contributing to a better understanding of the vascular components involved in this complex disease.

## 2. Materials and Methods

This was an observational, cross-sectional, non-interventional study conducted on 127 primary open-angle glaucoma (POAG) patients, 59 primary angle-closure glaucoma (PACG) patients, 34 pseudoexfoliative glaucoma (PEX) patients, and 55 normotensive glaucoma (NTG) patients (total of 275 glaucoma patients). The control group (CG, 92 patients) consisted of patients with cataract, refractive errors, and minor eyelids problems (verrucas, dermatochalasis). This study was conducted at the University Clinical Center of Serbia, University Eye Hospital, Belgrade.

Inclusion criteria for the subjects to enter this study were as follows:

For high-tension glaucoma: intraocular pressure (IOP) readings over 21 mmHg in repeated applanation tonometry (minimum 3), typical findings in gonioscopy (wide open angle in POAG, narrow or closed angle for PACG, high pigmentation in PEX), glaucomatous optic disk damage (cup/disk asymmetry between two eyes ≥0.2, neuroretinal rim thinning, notching, disk hemorrhage, or nerve fiber layer defect), and/or characteristic visual field defect.

For pseudoexfoliative glaucoma: the criteria listed above for open-angle glaucoma as well as the presence of typical pseudoexfoliative material.

For normotensive glaucoma: the criteria listed above for open-angle glaucoma, as well as IOP not exceeding 21 mmHg on repeated measurement.

The exclusion criterion was any type of preceding ophthalmic surgery.

A questionnaire administered by trained interviewers adhering to a standardized protocol was used to collect information on socio-demographic factors (age, self-reported sex), lifestyle factors (tobacco smoking, physical activity), medical history as advised by a physician (including cardiovascular and cerebrovascular diseases, diabetes, high cholesterol), current prescribed medications (including antihypertensives, diabetic and lipid-lowering). As arterial hyper- and hypotension, diabetes, carotid artery stenosis and migraine are potential confounder associated with glaucoma, we identified participants with these self-reported conditions. Cardiovascular diseases that were noted were arterial hypertension, angina, carotid artery stenosis, heart attack, heart failure, irregular heartbeat, intermittent claudication, and from cerebrovascular diseases migraine, ischaemic or haemorrhagic insult, and transitory ischaemic attack.

The blood pressure (BP) measurements were taken by a trained examiner based on a standard protocol—after quietly resting in a sitting position for 5 min, three consecutive blood pressure readings were obtained using a mercury sphygmomanometer [4]. The second and third readings were used to calculate the mean systolic blood pressure (SBP) and diastolic blood pressure (DBP) values for each participant.

ePWV was calculated by the formula that was recommended by the Reference Values for Arterial Stiffness Collaboration [5,6,7], from data on age and mean arterial BP. Mean arterial BP was calculated as diastolic BP + 0.4 × (systolic BP − diastolic BP).MBP = DBP + [0.4 × (SBP − DBP)](1)

ePWV was calculated based on the following equation:ePWV = 9.587 − 0.402 × age  +  4.560  ×  10^− 3^ × age^2^  − 2.621 × 10^− 5^ ×age^2^ × mean blood pressure (MBP)  +  3.176 × 10^− 3^ × age × MBP  −  1.832 × 10^− 2^ × MBP(2)

Statistical analysis was performed using SPSS program v.20 (IBM corp. released 2011). The normality of distribution of the study variables in particular groups was assessed with the Shapiro–Wilk test. The obtained results were processed by applying methods of descriptive (arithmetical mean, standard deviation) and analytical statistics, and between-group comparisons of tested variables were performed using ANOVA. A *p* value less than 0.05 was considered statistically significant.

This study was conducted in accordance with the Helsinki Declaration and approved by the local ethical committee and all patients gave informed written consent.

## 3. Results

Table 1 presents baseline characteristics of the participants overall (*n* = 367), data on lifestyle factors, medical history, current prescribed medications, blood pressure measurements, and ePWV values. All patients were matched by gender and age.

Statistically significant differences were found between patients with POAG and the CG (*p* = 0.042), and between those with NTG and the CG (*p* = 0.001). Results are shown in Table 2. Also, there was a statistically significant difference in ePWV values when comparing all tested patients with glaucoma and the control group (*p* = 0.001).

As for the potential confounders associated with glaucoma, a slight increase in the number of participants was registered in the global group with glaucoma (hypertension and diabetes), i.e., in testing between the groups with glaucoma, there were slightly more participants in the group with PEX, but without statistical significance (*p* = 0.79).

## 4. Discussion

The correlation between systemic blood pressure, the state of large blood vessels, intraocular pressure, and small blood vessels of the eye are determined by the anatomical and physiological characteristics of the eye. Aqueous humor originates from arterial blood, and its outflow also depends on the vascular system, as it is absorbed into the venous circulation. The influence of blood pressure on intraocular pressure has been evaluated in numerous studies. It is well known that low blood pressure is an important risk factor for normal-tension glaucoma progression, and on the other hand, systemic hypertension has been identified as a risk factor for glaucoma onset and progression. The first is considered to be a consequence of poor blood supply to the optic nerve, and the second is mainly due to an increase in ciliary blood flow and increased aqueous humor production on one hand and high episcleral venous pressure and a decrease in humor outflow on the other [8,9,10,11].

Arterial stiffness is an important parameter that reflects the state of blood vessels, primarily influenced by various factors, including aging, gender, hormonal influences, blood pressure, and genetic predispositions. The structural components of arterial walls—such as smooth muscle cells, elastin, and collagen—play significant roles in determining arterial stiffness. Considering the aspect of large blood vessels, arterial stiffness can be regarded the most important cause of increasing systolic and pulse pressure, but stiffening also affects smaller arterial blood vessels in all tissues [12]. Arterial stiffness is most often estimated on the basis of carotid–femoral pulse wave velocity (cfPWV) or brachial–ankle pulse wave velocity (baPWV) [13], although direct insight into the states of the walls of arterial blood vessels and their dynamic changes during each contraction of the heart muscle can also be obtained by other invasive and non-invasive methods [14]. ePWV is a marker that can indicate the risk of developing cardiovascular diseases, but lately this parameter is increasingly the focus of research in correlation with the risk of developing other diseases too. One of the biggest advantages of the ePWV index is that it is easily calculated from age and mean blood pressure (BP) using a recommended equation.

This study builds upon previous research [8] exploring the link between arterial stiffness and glaucoma, aiming to investigate whether distinct ePWV values exist among various glaucoma subtypes when compared with matched control subjects.

Our results demonstrate that there is a statistically significant difference in ePWV values when comparing all tested patients with glaucoma and the control group (*p* = 0.00). Additionally, further analysis reveals notable differences in ePWV between patients with POAG and NTG when contrasted with the control group, as indicated by ANOVA (POAG vs. control: *p* = 0.04; NTG vs. control: *p* = 0.00).

The relationship between arterial stiffness and glaucoma has previously been substantiated by findings from large-scale population studies. These investigations have shown that elevated brachial pulse pressure (beyond 70 mmHg) correlates with a 13% increased risk of developing POAG. Moreover, ePWV serves as a valuable predictor for glaucoma onset, demonstrating a positive association with future disease development. The link between increased arterial stiffness and small-vessel disease is well documented in brain tissue; similarly, it is plausible to apply this association to the optic nerve vasculature due to their shared embryonic origins and blood supply mechanisms [15]. There are few studies on the association between systemic arteriosclerosis and ocular hemodynamics in patients with glaucoma [16]. Supporting these ideas, a study by Shim et al. underscored the implications of systemic arterial stiffness in the development of glaucoma, particularly in diabetic populations, indicating that diabetes significantly heightens the risk of glaucoma when coupled with severe systemic arterial stiffness. This highlights the need for further investigation into the vascular factors contributing to glaucoma and the potential for ePWV as a biomarker for assessing risk and guiding management strategies in affected populations. It is unequivocally confirmed that diabetes is a risk factor for the occurrence of glaucoma if a person simultaneously has high-grade systemic arterial stiffness [17].

Based on our results, it seems that people with pseudoexfoliative glaucoma do not have a high epWV. This is consistent with the definition of PEX glaucoma as part of systemic basement membrane disease, in which vascular factors do not play a predominant role. The same applies to PACG patients, given that this type of glaucoma is a consequence of the anatomical characteristics of the anterior segment of the eye. We did not exclude subjects with cardiovascular diseases from our tested sample on purpose, because the population suffering from glaucoma is prone to systemic comorbidities and because we wanted to have a realistic population cross-section of conditions related to arterial stiffness.

Arterial stiffness is a measure of the elasticity of the arterial blood vessel wall. If arterial wall is softer (and its stiffness is low), it means that it will more easily comply with changes in blood pressure and the demands of the tissues to which the oxygenated blood is delivered. On the other hand, if the arterial stiffness is higher, much more power must be invested to deliver needed amount of oxygenated blood to the tissues; that is, the heart has to work harder. This will inevitably lead to an increase in blood pressure. For these reasons, it is important that arterial stiffness is not excessive. It is debatable whether it is good for arterial stiffness to be as low as possible, because with this, blood vessels can become too soft and will easily expand even at the slightest increase in blood pressure, which in certain situations can cause other unwanted events, such as hemorrhages. All of this applies to both large and small blood vessels because arterial stiffness represents a systemic condition of blood vessels throughout the body [18,19].

We are aware of the shortcomings of our study, the biggest of which can be considered the small sample size. Also, we did not perform a statistical analysis related to the stage of glaucoma and the level of arterial stiffness, because that part of this research is currently ongoing. Prospective studies on the possible relationship between arterial stiffness values and antiglaucoma therapy are also underway.

## 5. Conclusions

Overall, our research underscores the importance of investigating vascular factors in glaucoma and paves the way for future explorations that might better inform clinical practices and patient management. The findings of our study reinforce the importance of vascular health in understanding the complex etiology of glaucoma. By establishing a connection between ePWV and various types of glaucoma, this study contributes to a growing body of evidence indicating that monitoring arterial stiffness may be beneficial in the management and prevention of glaucoma, alongside traditional ocular assessments. Further studies are anticipated to expand on these findings and clarify the mechanisms at play, and there is a potential of ePWV to become a helpful tool in future risk assessment models.

## Figures and Tables

**Table 1 biomedicines-13-02033-t001:** Baseline characteristics, data on lifestyle and systemic conditions, ePWV values. HDL—high-density lipoprotein cholesterol; BP—blood pressure.

Number	Glaucoma	Control	POAG	PCAG	PEX	NTG
	275	92	127	59	34	55
Age	69 ± 9 (23–87)	66.4 ± 11 (44–85)	65.3 ± 11.5 (39–86)	66.4 ± 12.7 (37–85)	66.4 ± 16.5 (45–87)	61.6 ± 14.2 (23–87)
Female, %	52	63	58	54	44	60
Tobacco smoker, %	23	22	12	8	16	3
Physical activity, %	14	16	12	8	5	14
Diabetes, %	14	12	10	11	15	6
Cardiovascular disease, %	27	23	32	25	22	20
Cerebrovascular disease, %	3	1	2	0	1	0
Total cholesterol, mmol/L	5.12 ± 1.13	4.59 ± 1.09	4.78 ± 1.05	5.01 ± 1.11	5.09 ± 1.01	4.75 ± 1.04
HDL, mmol/L	1.53 ± 0.56	1.42 ± 0.44	1.44 ± 0.37	1.42 ± 0.41	1.41 ± 0.44	1.40 ± 0.39
Medications						
Antihipertensives	96	87	31	22	35	8
Diabetic medications	31	22	10	8	9	4
Lipid-lowering medications	22	20	6	6	8	2
Systolic BP, aortic, mmHg	128 ± 16	131 ± 18	130 ± 12	127 ± 10	132 ± 11	110 ± 16
Diastolic BP, aortic, mmHg	78 ± 9	74 ± 7	77 ± 9	74 ± 11	79 ± 9	67 ± 8
Heart rate, bpm	65 ± 9	64 ± 10	66 ± 12	65 ± 9	69 ± 9	65 ± 9
ePWV	11.247 ± 1.9	10.593 ± 2.1	11.141 ± 1.7	11.201 ± 1.1	11.007 ± 1.5	12.003 ± 1.7

**Table 2 biomedicines-13-02033-t002:** Comparison of ePWV values between the tested groups. CG—control group; POAG—primary open-angle glaucoma; PACG—primary angle-closure glaucoma; PEX—pseudoexfoliative glaucoma; NTG—normotensive glaucoma; ALL GL—all patients with glaucoma.

	CG	POAG	PACG	PEX	NTG
CG	/	0.042	0.251	0.542	0.001
POAG	0.042	/	0.623	0.665	0.134
PACG	0.251	0.623	/	0.463	0.194
PEX	0.542	0.665	0.463	/	0.174
NTG	0.001	0.134	0.194	0.174	/
ALL GL	0.001	/	/	/	/

## Data Availability

Research data are available on request.

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
