# Peer review of "Estimated Pulse Wave Velocity (ePWV) in Different Glaucoma Types"

_biomedicines, 2025, doi:10.3390/biomedicines13082033_

Round 1

Reviewer 1 Report

Comments and Suggestions for Authors

Thank you for the opportunity to review this interesting study. This topic is clinically important, but the manuscript would benefit from several key revisions. Please see more detailed comments below:

In the title and abstract section, the abstract mentions a statistically significant difference between all glaucoma patients and controls, but the reported p-value is 0.25, which is not significant (lines 23–25). This needs correcting and perhaps includes descriptive statistics for clarity. Additionally, the statement about implementing ePWV in clinical practice feels too strong, especially given the exploratory nature of your data. Consider rephrasing to something more cautious, such as: “may be a helpful tool in future risk assessment models.”.

In the Introduction, it would be stronger if you clearly explain how systemic vascular changes might influence the eye. Right now, the transition from general vascular dysfunction to glaucoma is a bit vague. (line 36-39).

In the methods section, It’s good that you excluded patients with prior eye surgeries, but what about systemic conditions like diabetes, hypertension, or cholesterol levels? These are all factors that affect arterial stiffness. (line 65). Additionally, the formula for ePWV is very hard to read and seems to contain some formatting errors (e.g., “z0.01x(age2)”). (line 75-77). Rewriting this with standard mathematical notation using latex, would really help the reader follow along. Furthermore, you mention using t-tests to compare groups. Given that you’re making several pairwise comparisons, a correction for multiple testing (like Bonferroni) would strengthen the statistical approach. Alternatively, ANOVA followed by post hoc testing may be more appropriate. (line 82).

In the results section, the claim that the difference between all glaucoma patients and the control group is statistically significant is again contradicted by the reported p-value (0.25). Please clarify. 

  • Multiple comparisons are made here using t-tests, but there’s no indication of whether any correction was applied. (Table 3).
  • P-value in Table 1 does not make sense. 

In the discussion section, some parts of the discussion feel more like a physiology lecture than a focused interpretation of your findings (line 120-130). Try to relate it directly back to the results of this study.

Not collecting data on things like smoking, BMI, or cholesterol is a notable limitation, I recommend listing these clearly in your limitations section.

The recommendation to start calculating ePWV in all glaucoma patients might be a bit premature. It’s a good idea, but perhaps frame it more as a direction for future research. 

Minor comments:

  • Use professional language throughout, some parts are difficult to understand.
  • Use standard Table labeling, there are awkward placement here and there such as P t-test in Table 1.
  • Use 3 decimal places for all p-values (follow standard reporting).

Look forward to receiving your revised manuscript.

Author Response

REPLY TO REVIEWER 1

Dear Reviewer,

Thank you for the time and effort you took to read and review our manuscript, and for your kind remarks.

  • In the title and abstract section, the abstract mentions a statistically significant difference between all glaucoma patients and controls, but the reported p-value is 0.25, which is not significant (lines 23–25). This needs correcting and perhaps includes descriptive statistics for clarity. Additionally, the statement about implementing ePWV in clinical practice feels too strong, especially given the exploratory nature of your data. Consider rephrasing to something more cautious, such as: “may be a helpful tool in future risk assessment models.”.

First remark about statistical significance is in place, and it was a typo, we have changed it (there is a statistical significant difference in ePWV values between all glaucoma patients and control group of 0.01)

In addition, we do agree that the statement that calculating ePWV should be a part of everyday clinical examination is too strong, and we did rephrase it to sound more realistic and cautious.

  • In the Introduction, it would be stronger if you clearly explain how systemic vascular changes might influence the eye. Right now, the transition from general vascular dysfunction to glaucoma is a bit vague. (line 36-39).

We accept the remark about the Introduction section, and we have added a more detailed explanation in the text.

  • In the methods section, It’s good that you excluded patients with prior eye surgeries, but what about systemic conditions like diabetes, hypertension, or cholesterol levels? These are all factors that affect arterial stiffness. (line 65). Additionally, the formula for ePWV is very hard to read and seems to contain some formatting errors (e.g., “z0.01x(age2)”). (line 75-77). Rewriting this with standard mathematical notation using latex, would really help the reader follow along. Furthermore, you mention using t-tests to compare groups. Given that you’re making several pairwise comparisons, a correction for multiple testing (like Bonferroni) would strengthen the statistical approach. Alternatively, ANOVA followed by post hoc testing may be more appropriate. (line 82).

We added data on systemic conditions of tested subjects (diabetes, hypertension and cholesterol levels, tobacco usage and else), as you suggested

We changed the way that formula for calculating ePWV is written, so that is easy to follow

We recalculated data of compared groups using ANOVA

  • In the results section, the claim that the difference between all glaucoma patients and the control group is statistically significant is again contradicted by the reported p-value (0.25). Please clarify. Multiple comparisons are made here using t-tests, but there’s no indication of whether any correction was applied. (Table 3).
  • P-value in Table 1 does not make sense. 

We have corrected an error related to the statistical significance of the difference between ePWV between groups, as you rightly noted

We changed the statistical analysis from t/test to ANOVA, and merged Tables

We have corrected the error of the p value in Table 2.

  • In the discussion section, some parts of the discussion feel more like a physiology lecture than a focused interpretation of your findings (line 120-130). Try to relate it directly back to the results of this study.
  • Not collecting data on things like smoking, BMI, or cholesterol is a notable limitation, I recommend listing these clearly in your limitations section.
  • The recommendation to start calculating ePWV in all glaucoma patients might be a bit premature. It’s a good idea, but perhaps frame it more as a direction for future research. 

Thank you for the comment on the discussion section, we accept the criticism, and have changed the text in hope to be more appropriate to the subject

As we have stated in the original version of the manuscript, this is a study that is still going on, and we added data on smoking, cholesterol, cardiovascular and cerebrovascular diseases in revised version of the manuscript.

We agree that we were carried away in the claim that the calculation of ePWV should be a part of the daily practice in the glaucoma clinic, so we changed that part of the text according to your suggestions.

  • Use professional language throughout, some parts are difficult to understand.
  • Use standard Table labeling, there are awkward placement here and there such as P t-test in Table 1.
  • Use 3 decimal places for all p-values (follow standard reporting).

We tried to bring the text closer to professional English. We corrected Tables, and added 3 decimal places for all p-values, as you have kindly suggested.

Reviewer 2 Report

Comments and Suggestions for Authors

This study compares the estimated pulse wave velocity (ePWV) among patients with different types of glaucoma, presenting a novel and clinically significant research topic.  The authors included 275 patients with various glaucoma subtypes (primary open angle glaucoma [POAG], primary angle closure glaucoma [PACG], pseudoexfoliative glaucoma [PEX], and normotensive  glaucoma [NTG]) and 92 control subjects with cataract. The findings fill a gap in understanding the relationship between different glaucoma subtypes and arterial stiffness, offering new insights into the vascular factors involved in glaucoma pathogenesis.

In terms of methodology and data analysis, the study adopts a single-center, cross-sectional observational design. The ePWV was calculated using a formula based on age and blood pressure, with standardized blood pressure measurement methods, demonstrating an overall reasonable design. Demographic characteristics were matched across groups to minimize confounding effects. Statistical analysis primarily employed t-tests to compare intergroup differences, which was appropriate. Considering the multiple group comparisons, further analysis using ANOVA could be explored.

The results show that ePWV was significantly higher in the POAG and NTG groups compared to the control group (p≈0.04 and p<0.001, respectively), while no statistically significant differences were observed in the PACG and PEX groups. This suggests that open angle and normotensive glaucoma patients may exhibit higher arterial stiffness, supporting the role of vascular factors in these subtypes. Conversely, the absence of elevated ePWV in PEX and angle closure glaucoma aligns with the limited contribution of vascular factors in their etiology. However, as this is a cross-sectional study, it can only reflect correlations rather than causality. The authors should avoid direct causal inferences in their conclusions and recommend further research to validate the clinical implications.

The study has some limitations. It did not account for other cardiovascular risk factors (e.g., dyslipidemia, diabetes) that might influence ePWV, potentially confounding the results. Incorporating multivariate analysis could enhance the reliability of the conclusions. Lastly, the results section states a p-value of 0.25 for "all glaucoma patients vs. controls," while Table 3 reports 0.01—an apparent discrepancy that should be corrected.

Author Response

REPLY TO REVIEWER 2

Dear Reviewer,

Thank you for taking the time to read our manuscript and for your kind criticism and suggestions how to improve it.

  • In terms of methodology and data analysis, the study adopts a single-center, cross-sectional observational design. The ePWV was calculated using a formula based on age and blood pressure, with standardized blood pressure measurement methods, demonstrating an overall reasonable design. Demographic characteristics were matched across groups to minimize confounding effects. Statistical analysis primarily employed t-tests to compare intergroup differences, which was appropriate. Considering the multiple group comparisons, further analysis using ANOVA could be explored.

We applied ANOVA statistical analysis, as you suggested, so the statistical data in the revised version of the paper were written based on that analysis.

  • The results show that ePWV was significantly higher in the POAG and NTG groups compared to the control group (p≈0.04 and p<0.001, respectively), while no statistically significant differences were observed in the PACG and PEX groups. This suggests that open angle and normotensive glaucoma patients may exhibit higher arterial stiffness, supporting the role of vascular factors in these subtypes. Conversely, the absence of elevated ePWV in PEX and angle closure glaucoma aligns with the limited contribution of vascular factors in their etiology. However, as this is a cross-sectional study, it can only reflect correlations rather than causality. The authors should avoid direct causal inferences in their conclusions and recommend further research to validate the clinical implications.

Thank you for the comment related to the interpretation of the results of our study, we tried to change that part of the text according to your suggestions, i.e. to avoid direct causal inferences in our conclusions.

  • The study has some limitations. It did not account for other cardiovascular risk factors (e.g., dyslipidemia, diabetes) that might influence ePWV, potentially confounding the results. Incorporating multivariate analysis could enhance the reliability of the conclusions. Lastly, the results section states a p-value of 0.25 for "all glaucoma patients vs. controls," while Table 3 reports 0.01—an apparent discrepancy that should be corrected.

Since we received objections regarding the data on the cardiovascular risks of the studied patients from two reviewers, and we have previously collected the given data, we processed them statistically and added them to the text. We believe this adds to the quality of this study.

As you rightly noticed, there was an error in the data in Table 3, which we corrected.

Reviewer 3 Report

Comments and Suggestions for Authors

Summary:

The study investigates the relationship between estimated pulse wave velocity (ePWV) – a new index for arterial stiffness found to be significant for mortality and cardiovascular risk – and various types of glaucoma. The paper compares subgroups within a cohort of 275 glaucoma patients diagnosed with POAG, PACG, PEX, and NTG, as well as 92 cataract control patients. The author reports that there was a statistically significant difference between the ePWV of the POAG and the controlled group, as well as the ePWV of the NTG and the controlled group; overall, there is also a significant difference between the ePWV of glaucoma patients versus the controlled group. The results of the study are congruent with the cause of distinct glaucoma types: typically, PACG and PEX are exhibited by patients with clear anatomical and systemic conditions that are distinct from vascular factors, while the manifestations of POAG and NTG are multifaceted and can be caused by vascular factors. Ultimately, the author demonstrated an explanable correlation between ePWV and glaucoma subtypes characterized by more complex and potential vascular-related manifestations. With its ease of measurement and calculator, ePWV could serve as a novel screening tool to identify patients at risk for glaucoma.

General Concept Comments:

- While the materials and methods specify that this is an observational, cross-sectional, non-interventional study, the paper does not provide the institution where the data comes from, nor the ethnic breakdowns. It would be helpful to have such patient information as the size of institution, the ethnic composition of the cohort, and the geographic location can influence the representativeness of the data.

- The central question of the paper is posed as “are arterial stiffness measured by ePWV associated with increased risk of developing glaucoma?”. In order to claim this causative relationship, the design technically would need to be a prospective study (following patients with and without arterial stiffness and seeing whether they develop glaucoma) rather than a cross-sectional study. With the current cross-sectional method, a more appropriate question is perhaps “Is there an association between arterial stiffness and glaucoma?”

- If the question is rephrased, the discussion should not mention that the study can “significantly help in identifying people who have a high risk of getting this disease” (line 201-202).

- It would be helpful for the authors to clarify why cataract patients, instead of healthy age-matched patients served as the control group in this study.

Specific Comments:

- While the t-test results are listed in Table 3, the actual ePWV values and their standard deviations for each glaucoma subcategories were not provided in the paper. Having these statistics is helpful for evaluating the direction and variability within and between each subgroups.

- The ePWV values provided for all glaucoma groups VS control group varies throughout the paper: Line 94 states that p = 0.25, table 2 states that p = 0.04, table 3 states that p = 0.01. Please provide consistent  values.

Comments on the Quality of English Language

sufficient

Author Response

REPLY TO REVIEWER 3

Dear reviewer,

Thank you for the effort and time allocated for the review of our manuscript. Thank you for the kind criticism.

  • While the materials and methods specify that this is an observational, cross-sectional, non-interventional study, the paper does not provide the institution where the data comes from, nor the ethnic breakdowns. It would be helpful to have such patient information as the size of institution, the ethnic composition of the cohort, and the geographic location can influence the representativeness of the data.

The institution where the study was conducted is not mentioned in the text, because according to the journal's propositions, all information that would reveal the author's identity should be avoided. However, we agree that ethnicity and geographic location are important for this type of study, so we added the data to the revised version of the paper.

  • The central question of the paper is posed as “are arterial stiffness measured by ePWV associated with increased risk of developing glaucoma?”. In order to claim this causative relationship, the design technically would need to be a prospective study (following patients with and without arterial stiffness and seeing whether they develop glaucoma) rather than a cross-sectional study. With the current cross-sectional method, a more appropriate question is perhaps “Is there an association between arterial stiffness and glaucoma?”
  • If the question is rephrased, the discussion should not mention that the study can “significantly help in identifying people who have a high risk of getting this disease” (line 201-202).

Thank you very much for pointing out that we need to reformulate the aim of our study according to the method we have applied. We have changed the text above according to your suggestions.

Once again, with the change of the question that is at the core of our study, we also changed the rest of the text, i.e. we removed the overly strong claim that calculating the ePWV index will help in identifying people who have a high risk of getting glaucoma.

  • It would be helpful for the authors to clarify why cataract patients, instead of healthy age-matched patients served as the control group in this study.

This study was performed on patients who came to the Clinic for Eye Diseases. The control group consisted of patients with cataracts, refractive errors and minor eyelids problems (verrucas, dermatochalasis). We stated this in new version of our manuscript.

  • While the t-test results are listed in Table 3, the actual ePWV values and their standard deviations for each glaucoma subcategories were not provided in the paper. Having these statistics is helpful for evaluating the direction and variability within and between each subgroups.

It was correctly noted that we failed to specify the real values of ePWV in the text and we added that data.

  • The ePWV values provided for all glaucoma groups VS control group varies throughout the paper: Line 94 states that p = 0.25, table 2 states that p = 0.04, table 3 states that p = 0.01. Please provide consistent  values.

We acknowledge that we made a typo related to the p value and have corrected it.

Round 2

Reviewer 1 Report

Comments and Suggestions for Authors

Thank you for addressing all my previous comments, I have no further feedback. All the best in the next stage of the publication journey.

Reviewer 2 Report

Comments and Suggestions for Authors

The authors present a cross-sectional observational study investigating the relationship between estimated pulse wave velocity (ePWV)—a surrogate marker of arterial stiffness—and different types of glaucoma. The manuscript addresses an understudied but clinically relevant topic, especially considering the increasing recognition of vascular contributions to glaucomatous optic neuropathy.
Notably, the authors have revised the manuscript to incorporate key reviewer suggestions from the previous round. They have:
Transitioned from simple t-tests to ANOVA for multiple-group comparisons, improving statistical rigor;
Added relevant cardiovascular comorbidity data (e.g., hypertension, diabetes) and accounted for their influence on ePWV;
Corrected previously inconsistent p-values in Table 3;
Revised their conclusion section to avoid direct causal language, aligning with the cross-sectional nature of the study.
These revisions considerably strengthen the overall quality and credibility of the paper.